# Switchable supramolecular helices for asymmetric stereodivergent catalysis

Ran Chen [1], Ahmad Hammoud[1], Paméla Aoun[1], Mayte A. Martínez-Aguirre [1], Nicolas Vanthuyne [2], Régina Maruchenko[1], Patrick Brocorens [3], Laurent Bouteiller[1] & Matthieu Raynal [1] ✉

Despite recent developments on the design of dynamic catalysts, none of them have been exploited for the in-situ control of multiple stereogenic centers in a single molecular scaffold. We report herein that it is possible to obtain in majority any amongst the four possible stereoisomers of an amino alcohol by means of a switchable asymmetric catalyst built on supramolecular helices. Hydrogen-bonded assemblies between a benzene-1,3,5-tricarboxamide (BTA) achiral phosphine ligand coordinated to copper and a chiral BTA comonomer are engaged in a copper-hydride catalyzed hydrosilylation and hydroamination cascade process. The nature of the product stereoisomer is related to the handedness of the helices and can thus be directed in a predictable way by changing the nature of the major enantiomer of the BTA comonomer present in the assemblies. The strategy allows all stereoisomers to be obtained one-pot with similar selectivities by conducting the cascade reaction in a concomitant manner, i.e. without inverting the handedness of the helices, or sequentially, i.e. by switching the handedness of the supramolecular helices between the hydrosilylation and hydroamination steps. Supramolecular helical catalysts appear as a unique and versatile platform to control the configuration of molecules or polymers embedding several stereogenic centers.

Molecular switches or non-covalent interactions have been combined with catalytic units in the same (macro)molecular scaffold to control the outcome of catalytic reactions[1–6]. Amongst this emerging area, selecting the configuration of the generated stereogenic element in a predictable manner is a current challenge that has been achieved by a limited variety of reconfigurable catalysts[7]. In the most-employed strategies, the intrinsically achiral catalytic unit is connected to a molecular[8–14], macromolecular[15,16], or supramolecular chiroptical switch[17–19], leading to pseudo-enantiomeric states upon activation of the switch[20] by a suitable stimulus (Fig. 1a). These switchable asymmetric catalysts have been exclusively employed for the generation of chiral molecules having opposite configurations, i.e. enantiodivergency. Alternative strategies have been pursued by the Leigh

group[21–23] for which two pseudo-enantiomeric catalysts are present from the beginning in the reaction mixture. This yields an elegant example of stereodivergency in which a stoichiometric amount of a substrate anchored to a molecular machine is sequentially transformed by the catalyst pseudo-enantiomers (Fig. 1b)[22]. In a different design, the simultaneous operation of a pair of enantioselective switchable catalysts was prevented by mutual inhibition[21]. Light[11–13,24–26], redox potential[9,27], and chemical triggers[17,28–31] are the stimuli that hold most promise for interconverting the enantiomeric states of a catalyst on a timescale that is compatible with a chemical process, with the ultimate goal of controlling multiple stereogenic elements in small molecules, i.e. stereodivergency[32], or during a polymerization[33]. However, these systems must overcome significant

[1]Sorbonne Université, CNRS, Institut Parisien de Chimie Moléculaire, Equipe Chimie des Polymères, 4 Place Jussieu, 75005 Paris, France. [2]Aix Marseille Université, Centrale Marseille, CNRS, iSm2, UMR 7313, 13397 Marseille, Cedex 20France. [3]Service de Chimie des Matériaux Nouveaux, Institut de Recherche sur les Matériaux, Université de Mons, 20B-7000, 20 B-7000 Mons, Belgium. ✉e-mail: matthieu.raynal@sorbonne-universite.fr

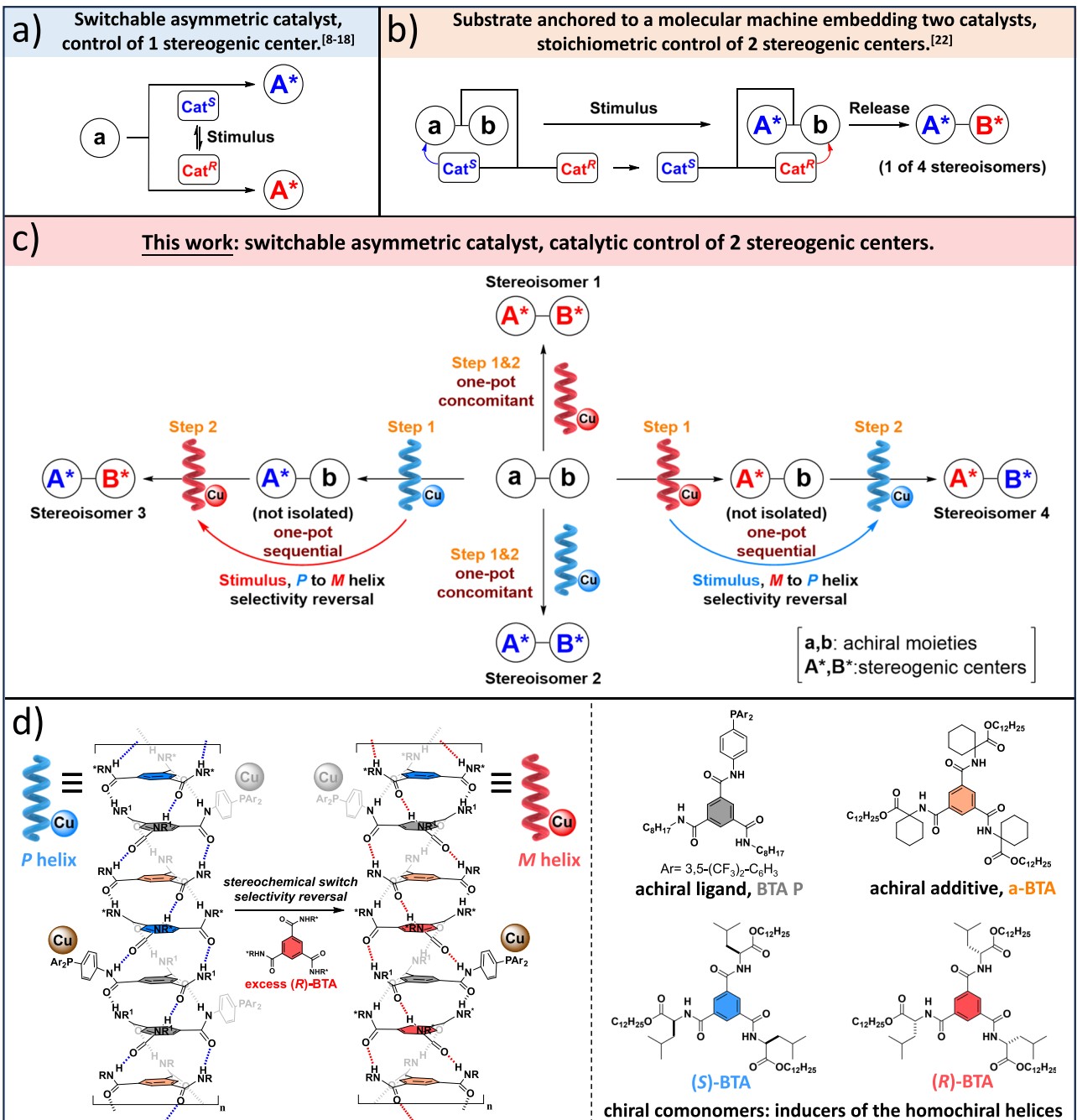

**Fig. 1 | Switchable asymmetric catalysis for enantio- and stereodivergency.**
**a** Reconfigurable asymmetric catalysts based on switchable asymmetric catalysts (generation of the enantiomers of A* from the same substrate a)[8–19].
**b** Stoichiometric control of two stereogenic centers A* and B* by means of a substrate a-b anchored to a molecular machine embedding two complementary catalysts[22]. **c** Schematic representation of the concept presented in this work. A stereochemically switchable helical catalyst is used for the catalytic control of two stereogenic centers A* and B* from the same molecular scaffold a-b. The supramolecular helices in each step embed the same achiral phosphine copper complex but differ by the major enantiomer of the BTA comonomer as indicated in (**d**).
**d** Left: schematic representation of the supramolecular helical BTA copolymers used in this study and of the conversion between their right-handed and left-handed states leading to selectivity reversal in the catalytic reaction. Right: molecular structures of the corresponding BTA monomers.

challenges: i) the stimulus and the substrate(s)/reactant(s)[34], or catalyst must be compatible[35], ii) the catalyst pseudo-enantiomers must provide perfectly opposite selectivities[24], and iii) the stereochemical switch must be rapid on the reaction timescale[12,36]. The implementation of a switchable asymmetric catalyst for the in-situ control of multiple stereogenic centers in a single molecular scaffold is an unmet challenge to date.

Recently-developed helical catalysts[37], with intrinsically achiral metal centers anchored at the periphery of covalent[38–51] or supramolecular helices[17,52–59], have particular advantages to reach this goal. For supramolecular helices, a chiral comonomer[40] is used to control the sense of rotation of the hydrogen-bonded assemblies supporting the catalytic centers. First, the direction of the asymmetric reaction is imposed by the handedness of the helices, making possible to control the direction of the catalytic process towards each enantiomer of a product. Importantly, these helices are dynamically chiral which make possible to fully interconvert their sense of rotation according to the nature of the major chiral monomer present in their

backbone. Second, well-established "sergeants-and-soldiers" and "diluted majority rule" principles are operative in these systems allowing homochiral helices to be obtained even if composed of a little amount of chiral monomers or of a scalemic mixture of monomer enantiomers, respectively[60,61]. The implementation of these unique "sergeants-and-soldiers" and "diluted majority rule" phenomena in catalysis enable optimal selectivities to be reached for both directions by simply tuning the supramolecular polymer composition. Third, the supramolecular helices are intrinsically dynamic: an important feature for switching the helix handedness and thus inverting the selectivity in situ.

We previously demonstrated that helical hydrogen-bonded stacks composed of a benzene-1,3,5-tricarboxamide[62,63] (BTA) achiral phosphine ligand and a small amount of enantiopure monomers provide good enantioselectivities in the copper-catalyzed hydrosilylation of 4-nitroacetophenone[56] and in the copper-catalyzed hydroamination of styrene[58]. In addition, changing the nature of the major BTA enantiomer afforded interconversion between enantiomeric helical catalysts within seconds[17]. The stereochemical switch was simply achieved by adding an excess of a chiral comonomer that acted as a chemical stimulus to invert the selectivity of the catalyst (as represented by the addition of the red-colored monomer in Fig. 1d). However, these previous results do not exploit the possibility to control the configuration of two stereogenic centers within the same molecular scaffold. Thanks to the unique development of these BTA helical catalysts, we report herein that any amongst the four possible stereoisomers of an amino alcohol can be obtained in majority through a one-pot hydrosilylation/hydroamination cascade reaction conducted either in a concomitant manner, i.e. without inverting the handedness of the helices, or sequentially, i.e. by switching the handedness of the supramolecular helices between the hydrosilylation and hydroamination steps (Fig. 1c).

## Results

### Structure of the BTA helical catalyst

We selected BTA monomers that previously proved to be suitable for the copper-catalyzed hydroamination of styrene (Fig. 1d)[58]. First component is **BTA P**, an achiral ligand with $CF_3$ groups at the *meta* positions of two of the aryl moieties connected to the phosphorus atom that were beneficial for both activity and enantioselectivity. Second component is **(S)-BTA** or **(R)-BTA**, chiral BTA comonomers derived from Leucine, that efficiently intercalate into the stacks of **BTA P**[57] to generate right-handed or left-handed helices, respectively, and in turn provide optimal enantioselectivities in the hydrosilylation (HS) and hydroamination (HA) catalytic processes. Third component is **a-BTA**, an achiral additive, that was found to greatly improve the magnitude of the "sergeants-and-soldiers" and "diluted majority rule" effects in catalytic coassemblies[56,64]. In other words, **a-BTA** allows to decrease the amount and optical purity of the chiral monomer while maintaining single-handed helices. For example, in the HA of styrene, an optimal selectivity of ca. 80% ee (enantiomeric excess) for the amine product was obtained with as low as 2.5% of **(S)-BTA** in the coassemblies, which represent one chiral monomer for 20 copper catalytic centers[58]. In addition, **a-BTA** was found to increase the yield and optimal selectivity of the reaction. All these monomers assemble into helical copolymers upon mixing in toluene, as schematically represented in Fig. 1d and previously characterized[58]. To implement the concept presented in Fig. 1c, we will conduct four cascade reactions: (i) two reactions with helical terpolymers of fixed composition for both steps of the catalytic process, hence embedding **BTA P** (coordinated to copper), **a-BTA** and pure **(S)-BTA** (or **(R)-BTA**) [concomitant process, no switch of the catalyst handedness, no selectivity reversal], and (ii) two reactions with helical polymers having different compositions for each step of the catalytic process, i.e. a helical terpolymer for the first step as above and a tetrapolymer for the second step which contains a scalemic mixture of the chiral monomers

[sequential process, switch of the catalyst handedness, selectivity reversal]. In the latter case, the switch is achieved by adding an excess of the chiral monomer initially in minority in the helical terpolymer, thus generating single-handed helices of opposite handedness thanks to the dynamic nature of the helices and the efficiency of the "diluted majority rule" effect[56].

### Cascade reaction without switch of the catalyst handedness [concomitant process]

Building on our previous results with copper-functionalized BTA helices[56,58] and inspired by the possibility to engage Cu-H catalysts in sequential transformations[65–68], we selected 3-vinylacetophenone (**VPnone**) as the substrate to implement our concept. Preliminary experiments support that the selectivity of the reaction can be controlled solely by the catalyst, not the substrate, an important parameter to obtain all the stereoisomers with similar selectivities (see Supplementary Figs. 10–11). Initial tests have been conducted under conditions similar to those established for the HA of styrene[58]. More precisely, the fraction of chiral comonomer ($f_s$) over all BTA monomers is fixed to 20%. Isothermal Titration Calorimetry (ITC) experiments reveal that this supramolecular helical catalyst is stable above 0.4 mM at 293 K and 1.3 mM at 313 K in toluene: the concentration in **BTA P** has thus been set to 11 mM to ensure that helices are maintained during the full catalytic process conducted at 313 K (Supplementary Fig. 2). In addition, an excess of amine electrophile and silane is used to enhance the yield of the reaction, and the catalytic screening is performed without exclusion of air[58]. In this part, silane and amine electrophile are engaged from the beginning of the reaction, thereby enabling both HS and HA reactions to start concomitantly (see Note 1 in the Methods section).

With these initial conditions in hands, and **(S)-BTA** as the chiral comonomer, the expected product 1-[3-(1-dibenzylaminoethyl)]-acetophenol, **APnol**, is obtained in ca. 66% yield, and **(R,S)-APnol** is the main stereoisomer (97% ee, diastereomeric ratio of 3.6:1). **APnol** is formed together with a small amount of **EPnol** (ca. 10%) and 3-vinylacetophenol (**VPnol**, 14%, see the structures in Table 1). The former probably comes from the protonation of the alkyl copper catalytic species by residual water[69,70] whilst the latter indicates incomplete HA of the vinyl function, a point confirmed by NMR monitoring of the concomitant process (Supplementary Fig. 3d). Thorough screening of various parameters established that addition of one equivalent of tris[3,5-bis(trifluoromethyl)phenyl]phosphine, $(P(3,5-(CF_3)_2-C_6H_3)_3)$, relatively to copper is beneficial for the yield of the reaction (Supplementary Table 1). The addition of a secondary ligand is a common strategy used to enhance the performance of Cu-H type catalysts[71,72]. Other aromatic solvents were also probed but they showed no obvious advantage relatively to toluene (Supplementary Table 2). **APnol** is now obtained in 76% yield and this constitutes our optimized conditions for the concomitant process (Table 1, entry 1).

Engaging **(R)-BTA** instead of **(S)-BTA** in the catalytic mixture leads to **(S,R)-APnol** with similar yield and selectivity as expected for homochiral BTA helical catalysts adopting opposite screw-sense preferences (Table 1, entry 2). Control experiment indicates that the role of **a-BTA** in these BTA helical catalysts for the cascade reaction appears to be similar to that observed previously in the HA of styrene i.e. it increases the yield and selectivity of the HA step (Supplementary Table 1). Selecting the optimized conditions of Table 1, the concomitant HS/HA reaction can be performed on 1 mmol scale yielding **APnol** in ca. 50% isolated yield and similar selectivity (Supplementary Table 3).

### Selection of the conditions for the sequential reaction

Two prerequisites are needed in order to control the configuration of the stereogenic centers formed by sequential transformations involving a switchable asymmetric catalyst: i) different reaction rates for the

**Table 1 | Catalytic results for the asymmetric HS/HA cascade reaction of VPnone [concomitant reactions]**

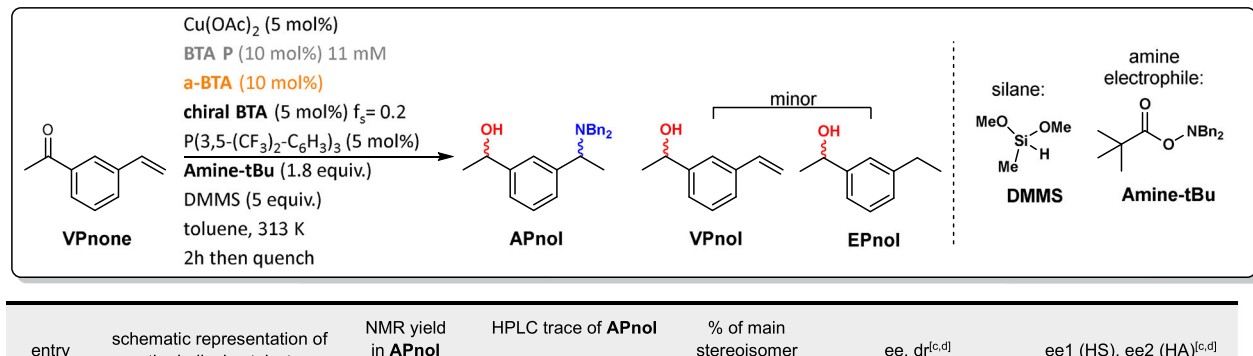

| entry | schematic representation of the helical catalyst | NMR yield in **APnol** (±5%)[b] | HPLC trace of **APnol** $(R,R)$ $(S,R)$ $(S,S)$ $(R,S)$ | % of main stereoisomer of **APnol**[c,d] | ee, dr[c,d] | ee1 (HS), ee2 (HA)[c,d] |
|---|---|---|---|---|---|---|
| 1 | | 76% | | 78% $(R,S)$ | 96% $(R,S)$, 3.8:1 | 69%, 84% |
| 2 | | 78% | | 75% $(S,R)$ | -96% $(S,R)$, 3.3:1 | -64%, -83% |

BTA helical catalysts have the same handedness for both hydrosilylation (HS) and hydroamination (HA) reactions yielding **(R,S)-APnol** and **(S,R)-APnol** as the main stereoisomers when **(S)-BTA** (blue helix) and **(R)-BTA** (red helix) are used as the chiral monomers, respectively.[a] DMMS: dimethoxymethylsilane. ee: enantiomeric excess. dr: diastereomeric ratio.
[a] $f_s = [(S)\text{-}BTA]/([(S)\text{-}BTA] + [BTA\ P]+[a\text{-}BTA])$ or $f_s = [(R)\text{-}BTA]/([(R)\text{-}BTA] + [BTA\ P]+[a\text{-}BTA])$. See Supplementary Fig. 3 and Supplementary Tables 1-3 for more details.
[b] An internal standard (1,3,5-trimethoxybenzene) is used to measure the NMR yield in **APnol** on the crude sample.
[c] Error bars: % of main stereoisomer ±1%, ±1% ee, ±0.2 dr, ±1% ee1, ±1% ee2 when **(S,R)-APnol** is the main stereoisomer, % of main stereoisomer ±2%, ±1% ee, ±0.4 dr, ±3% ee1, ±1% ee2 when **(R,S)-APnol** is the main stereoisomer. Uncertainties are related to the integration of the HPLC signals (see Methods). The absolute and relative configurations are established according to experiments conducted with **DTBM-SEGPHOS** as ligand and modelling studies (see Supplementary Figs. 16–19).
[d] The enantiomeric excess and diastereomeric ratio are obtained from the chiral HPLC analyses of the crude samples. Ee1 (positive for the (R)-enantiomer of the alcohol) and ee2 (positive for (S)-enantiomer of the amine) are extracted from the HPLC traces as indicated in Supplementary Fig. 20 and this convention was followed throughout this paper.

two catalytic transformations, and ii) rapid and full inversion of the enantiomeric state of the catalyst. Hydrosilylation and hydroamination reactions both require a silane reagent to proceed but HA require an additional amine electrophile: it allows HS and HA steps to be performed fully independently by adding the amine electrophile after completion of the HS step. We thus monitored the consumption of **VPnone** under the optimized conditions reported in Table 1 as well as under other conditions. We found out that the rate of the hydrosilylation reaction varies depending on the conditions, but not the selectivity (63–70% ee in **VPnol**, Supplementary Figs. 4 and 5). It means that the time for the HS step must be adapted for the sequential reactions (vide infra).

We next probed the possibility to invert the handedness of the BTA helical catalyst under conditions similar to those used to perform the catalytic reaction. A CD spectrum of a solution containing the pre-catalytic system, i.e. all BTA monomers, copper acetate, and the secondary phosphine ligand, was recorded at 313 K. An excess of **(R)-BTA** was then added to this solution leading to 50% ee in favour of the **(R)-BTA** monomers in the supramolecular tetrapolymer. The solution was stirred for 2 minutes and analyzed by CD. CD spectra before and after addition of **(R)-BTA** are mirror images (Fig. 2). These CD spectra exhibit a main CD band which belongs exclusively to **BTA P** since it is the only BTA monomer that absorbs in that region. We previously correlated this induced CD band to the selectivity observed in the copper-catalyzed HS reactions, i.e. the enantioselectivity (up to optimal selectivity that can be obtained by the system under study) is proportional to the intensity of this CD band[57,59]. As the present BTA coassemblies contain a sufficient number of enantiopure BTA monomers to be homochiral[56,58], the inverted CD signals indicate that all **BTA P** monomers coordinated to copper are positioned in opposite chiral environments as the result of a full stereochemical inversion of the handedness of the helical assemblies (from right-handed to left-handed) upon addition of **(R)-BTA** monomers. CD spectra of the

supramolecular tetrapolymer upon formation from preexisting terpolymer or after cooling from the monomeric state are virtually identical thus indicating that the thermodynamic state is reached after switching of the helix handedness (Supplementary Fig. 6).

These CD analyses corroborate that **(R)-BTA** monomers can play the role of the stereochemical triggers for inverting the enantiomeric preference of the dynamic helical catalyst in situ. The exact time needed to achieve this stereochemical switch has not been determined precisely but three minutes of mixing appears reasonable to ensure full stereochemical inversion and to limit side reaction at the vinyl function.

**Cascade reaction with switch of the catalyst handedness [sequential process]**

We then performed each step of the sequential process with the BTA helical catalyst having opposite handednesses in the same conditions as the previous CD experiment (Fig. 2). Even though **APnol** is obtained in low yield under these conditions (20-30%), we were pleased to see that the expected stereoisomers are now the dominant species. More precisely, the (R,R) and (S,S) stereoisomers of **APnol** are the major stereoisomers when the handedness of the catalyst is switched from right to left and from left to right before the HA step, respectively (Table 2, entries 1 and 2). Enantiomeric excesses for each step have opposite signs as expected for reactions conducted with a catalyst displaying opposite intrinsic enantioselectivities. However, the values of ee2 in entries 1 and 2 indicate that the selectivity of the HA step is not optimal, thus lowering the overall selectivity of the sequential reaction. This is likely due to the fact that the stereochemical switch was not complete when **Amine-tBu** was added and thus a small fraction of the silyl ether of **VPnol** (estimated to ca. 20%, see Note 2 in the Methods section) was converted by catalytic helices (or helical fragments) of the unwanted, non-switched handedness (see Note 3 in the Methods section).

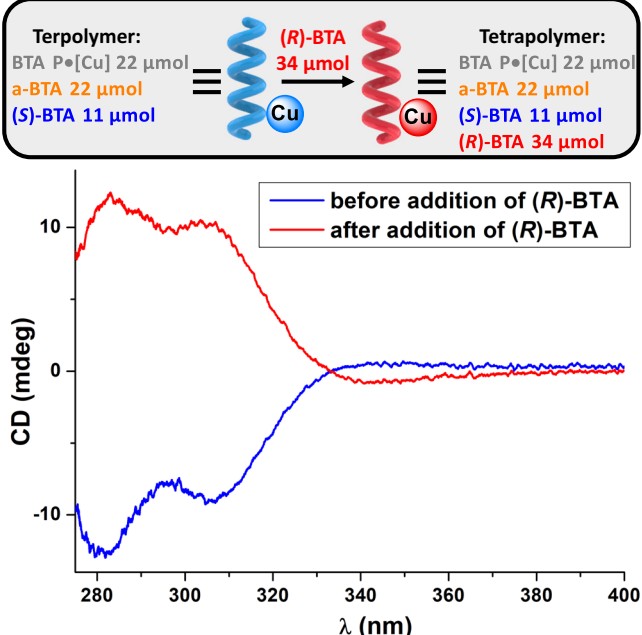

**Fig. 2 | Switching of the handedness of the BTA helical precatalysts.** CD spectra of the pre-catalytic mixture before and after addition of (**R**)-BTA (toluene, 313 K). BTA P•[Cu]= **BTA P** (22 μmol), Cu(OAc) (11 μmol), and P(3,5-(CF₃)₂-C₆H₃)₃ (11 μmol). After addition of (**R**)-BTA (34 μmol) dissolved in a minimal amount of toluene, the helical coassemblies contain 45 μmol of enantiopure monomers ($f_s$= 0.5) with a 50% ee in favour of (**R**)-BTA. A very small CD signal is detected at ca. 350 nm.

This negative effect was ultimately circumvented by lowering the amount of **a-BTA** to 5 mol% (Table 2, entries 3 and 4), which probably leads to slightly shorter and more dynamic terpolymers as a consequence of the decrease of the total BTA monomer concentration. The (R,R) and (S,S) stereoisomers of **APnol** are now obtained with a selectivity that matches the one obtained for the concomitant process: ca. 70% and 78% of the main stereoisomer of **APnol** for the sequential and concomitant processes, respectively (compared data in Table 1 and entries 3-4 in Table 2). Note that the amount of copper catalyst used in the concomitant and sequential processes is identical. Likewise, the fact that all **APnol** stereoisomers are obtained with similar selectivities discard any potential chiral match or mismatch between the intermediate **VPnol** and the helical catalyst and demonstrate the feasibility of the concept present in Fig. 1c, d.

Having established that a full inversion of the catalyst handedness can be achieved under air in the conditions similar to those of Fig. 2, we next probe the possibility to increase the overall yield of the sequential process. Analyses of the reaction mixtures of Table 2, entries 3 and 4 and a control experiment with a sequential reaction performed without addition of the chiral BTA monomers suggest that the low yield is related to catalyst deactivation (Supplementary Figs. 8 and 9). We thus hypothesized that O₂-free and anhydrous conditions may improve the efficiency of the sequential reaction. The reaction times were adapted since it appears that the rate of the HS step is dramatically decreased under these conditions (Supplementary Fig. 5). **APnol** is now formed in ca. 65% NMR yield and can be isolated in ca. 40% yield, close to the isolated yield obtained in concomitant processes, confirming that by-products are significantly minimized under these conditions (Table 2, entries 5 and 6, and Supplementary Fig. 9). The (R,R) and (S,S) stereoisomers of **APnol** are obtained as the main stereoisomers as expected for each step being promoted by a helical catalyst with opposite handedness, but with a lower selectivity (ca. 80% ee, dr= 1.3:1) compared to the same reaction performed under air. Examining the enantiomeric excess of each step reveals that the origin of the

lower selectivity comes from the HS step, which is far from being optimal, indicating that the HS of **VPnone** is not completed before the stereochemical switch (see Note 3 in the Methods section). In contrast, the selectivity of the HA step is of ca. 70% ee, with opposite sign compared to ee1, inferring that full stereochemical switch occurs under these conditions. Attempts to improve further the selectivity of the sequential reaction performed under nitrogen and anhydrous conditions towards the (R,R) and (S,S) stereoisomers of **APnol** were not successful since a delicate balance between catalyst activity (for HS step) and catalyst dynamicity (for the stereochemical switch) was not reached (Supplementary Table 4). Data shown in Table 2 and Supplementary Table 4 nevertheless represent unique examples of reactions for which the stereoisomer(s) that require(s) opposite catalyst selectivities is(are) formed one pot with an ee> 76% by means of a single catalytic system.

## Discussion

The four stereoisomers of **APnol** can thus be obtained with similar selectivities either from a concomitant process in which both steps are performed with a left-handed or right-handed helical catalyst or sequential reactions (under air) in which the handedness of the BTA helix is fully switched in between the HS and HA steps (Fig. 3). Dual catalysis[32] and cascade catalysis[65] are the most efficient strategies to achieve asymmetric stereodivergency with molecular catalysts but require several catalysts and/or multi-pot procedures because of the non-dynamic nature of the catalysts engaged in these reactions. The present work represents an alternative method to achieve stereodivergency in which all stereoisomers can be obtained one-pot thanks to a switchable asymmetric catalyst. It is also notable that because of the efficient control of the chiral and dynamic properties of the helical coassemblies, asymmetric stereodivergency is achieved with an achiral ligand thereby representing a powerful implementation of artificial chirogenesis[73].

In summary, the present work demonstrates the possibility to achieve asymmetric stereodivergency by means of a switchable asymmetric copper catalyst engaged in two transformations. The four stereoisomers of an amino alcohol are accessible one-pot, thus avoiding the isolation of the product intermediate from the initial catalytic system that is required when two enantiomers of a non-switchable catalyst are needed for each step of a sequential catalytic process. The approach is made possible by the incorporation of an achiral benzene-1,3,5-tricarboxamide (BTA) phosphine ligand in a supramolecular polymer for which the optical purity and handedness are controlled by an enantiopure BTA monomer derived from Leucine. The work benefits from the fact that the direction of the asymmetric reaction is directly related to the handedness of the helical polymer, which can be switched in two opposite directions by selecting the main BTA enantiomer in the polymers at each step of the reaction, leading to predictable configuration of two stereogenic centers. The stereochemical switch occurs through the addition of an excess of the BTA enantiomer acting as a chiral chemical trigger that inverts the stereochemical preference of the copper catalyst. The achiral BTA additive present in the supramolecular terpolymer, as well as the secondary phosphine ligand, improve the selectivity and yield of the reaction, respectively. Not only a fine tuning of the chirality of the supramolecular assemblies but also a proper control of their dynamicity is key to address stereodivergency. The present work demonstrates the feasibility of the concept to select one major (70%-79%) amongst four possible stereoisomers of an amino alcohol by applying the supramolecular helical catalyst in either concomitant (with no inversion of catalyst handedness) or sequential (with inversion of catalyst handedness) hydrosilylation and hydroamination reactions. The concept has been demonstrated for copper hydride type catalysis but can be reasonably extended to other catalytic processes and to the control of more than two stereogenic centers in small molecules or polymers[33]. It

**Table 2 | Catalytic results for the implementation of the concept in the asymmetric HS/HA cascade reaction of VPnone [sequential reactions]**

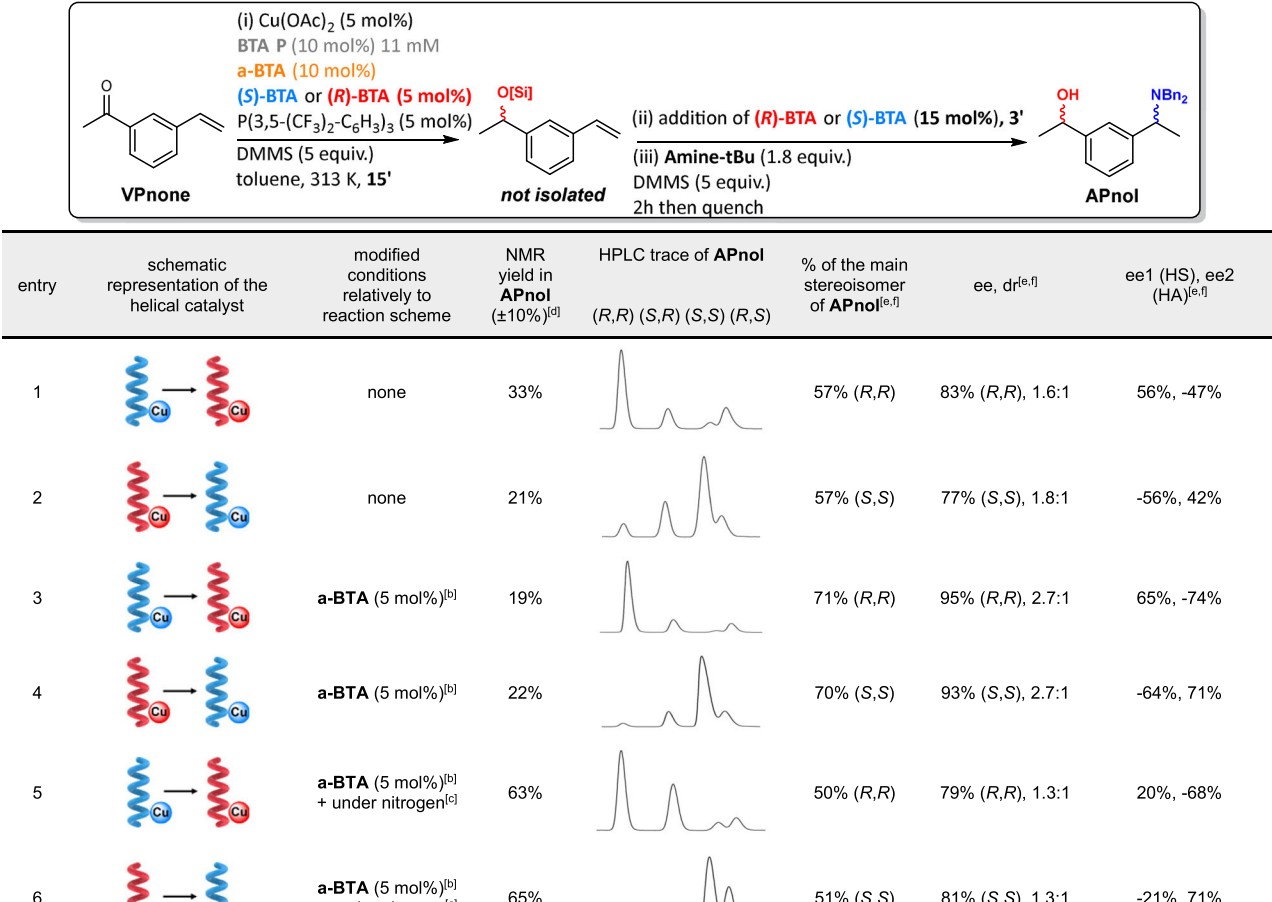

| entry | schematic representation of the helical catalyst | modified conditions relatively to reaction scheme | NMR yield in **APnol** (±10%)[d] | HPLC trace of **APnol** (R,R) (S,R) (S,S) (R,S) | % of the main stereoisomer of **APnol**[e,f] | ee, dr[e,f] | ee1 (HS), ee2 (HA)[e,f] |
|---|---|---|---|---|---|---|---|
| 1 | | none | 33% | | 57% (R,R) | 83% (R,R), 1.6:1 | 56%, -47% |
| 2 | | none | 21% | | 57% (S,S) | 77% (S,S), 1.8:1 | -56%, 42% |
| 3 | | a-BTA (5 mol%)[b] | 19% | | 71% (R,R) | 95% (R,R), 2.7:1 | 65%, -74% |
| 4 | | a-BTA (5 mol%)[b] | 22% | | 70% (S,S) | 93% (S,S), 2.7:1 | -64%, 71% |
| 5 | | a-BTA (5 mol%)[b] + under nitrogen[c] | 63% | | 50% (R,R) | 79% (R,R), 1.3:1 | 20%, -68% |
| 6 | | a-BTA (5 mol%)[b] + under nitrogen[c] | 65% | | 51% (S,S) | 81% (S,S), 1.3:1 | -21%, 71% |

BTA helical catalysts have opposite handedness for the hydrosilylation (HS) and hydroamination (HA) steps yielding **(R,R)-APnol** and **(S,S)-APnol** as the main stereoisomers when an excess of **(R)-BTA** (blue to red helix) or **(S)-BTA** (red to blue helix) are added to the catalytic mixture, respectively.[a] See Supplementary Figs. 7-9 and Supplementary Table 4 for more details. DMMS: dimethoxymethylsilane. ee: enantiomeric excess. dr: diastereomeric ratio.

[a]The reactions are performed without exclusion of air, expect otherwise stated.

[b]The amount of **a-BTA** is of 5 mol% instead of 10 mol%.

[c]The reaction is performed under nitrogen. The reaction times are adapted as follows: 60 minutes and overnight for the HS and HA steps, respectively. The reactions reported in entries 5 and 6 are performed on a ca. 1.0 mmol scale leading to **APnol** in 38% and 40% isolated yields, respectively.

[d]An internal standard (1,3,5-trimethoxybenzene) is used to measure the yield in **APnol** on the crude sample.

[e]Error bars: % of main stereoisomer ±1%, ±1% ee, ±0.2 dr, ±1% ee1, ±1% ee2 when (R,R)-**APnol** is the main stereoisomer, % of main stereoisomer ±2%, ±1% ee, ±0.4 dr, ±3% ee1, ±1% ee2 when (S,S)-**APnol** is the main stereoisomer.

[f]The enantiomeric excess and diastereomeric ratio are obtained from the chiral HPLC analyses of the crude (entries 1–4) or the purified (entries 5–6) samples.

is also conceivable that the catalytic properties and the dynamicity of this class of supramolecular helical catalysts can be improved by a proper design of the molecular structure of the monomers. Work along this direction is currently underway in our laboratory.

## Methods
### Synthetic procedures
The syntheses of **BTA P, a-BTA, (S)-BTA, (R)-BTA, Amine-DM** and **Amine-tBu** were reported previously[56,58]. **(S)-BTA** and **(R)-BTA** monomers used in this study have been purified by preparative HPLC. **VPnone** was prepared adapting a literature procedure (Supplementary Figs. 47 and 48)[74]. (S)- and (R)-DTBM-SEPHOS ( > 99%, TCI), PdCl₂ (99%, Sigma-Aldrich), PPh₃ (99%, Sigma-Aldrich), P(4-CH₃-C₆H₄)₃ (98%, Sigma-Aldrich), P(3,5-(CH₃)₂-C₆H₃)₃ (96%, Sigma-Aldrich), tBuPPh₂ (97%, Sigma-Aldrich), 1,2-bis(diphenylphosphino)ethane ( > 97%, TCI), 1,2-bis(diphenylphosphino)benzene ( > 98%, TCI), Cu(OAc)₂ (98%, Sigma-Aldrich), P(3,5-(CF₃)₂-C₆H₃)₃ (97%, ABCR), P(3,5-(CH₃)₂-4-OCH₃-C₆H₂)₃ (97%, Sigma-Aldrich), DMMS (97%, Alfa Aesar), 3'-

bromoacetophenone ( > 98%, TCI), Cs₂CO₃ (>98%, TCI) and potassium vinyltrifluoroborate (95%, Sigma-Aldrich) were purchased and used without any purification. Deuterated solvents were purchased from Eurisotop and used without further purification. Anhydrous solvents were obtained from a solvent purification system (IT-Inc). Purification by flash chromatography was performed by adsorbing the samples on silica; the adsorbed samples were introduced in the solid loader and purified by means of Reveleris X2 purification system (Buchi®) using pre-packed silica cartridges Ecoflex® (irregular 50 μm silica) of 25 g (for ca. 1 mmol catalytic samples). ¹H NMR spectra were recorded on a Brucker 400 AVANCE or 300 AVANCE (400 and 300 MHz respectively) and are calibrated with residual CDCl₃ protons signals at δ7.26 ppm. ¹³C NMR spectra were recorded on a Brucker 400 AVANCE or 300 AVANCE (100 and 75 MHz respectively) and are calibrated with CDCl₃ signal at δ77.16 ppm. Data are reported as follows: chemical shift (δ ppm), multiplicity (s= singlet, d= doublet, t= triplet, q= quartet, dd= doublet of doublets, dt= doublet of triplets, m= multiplet, bs= broad signal), coupling constant (Hz) and integration. Exact mass

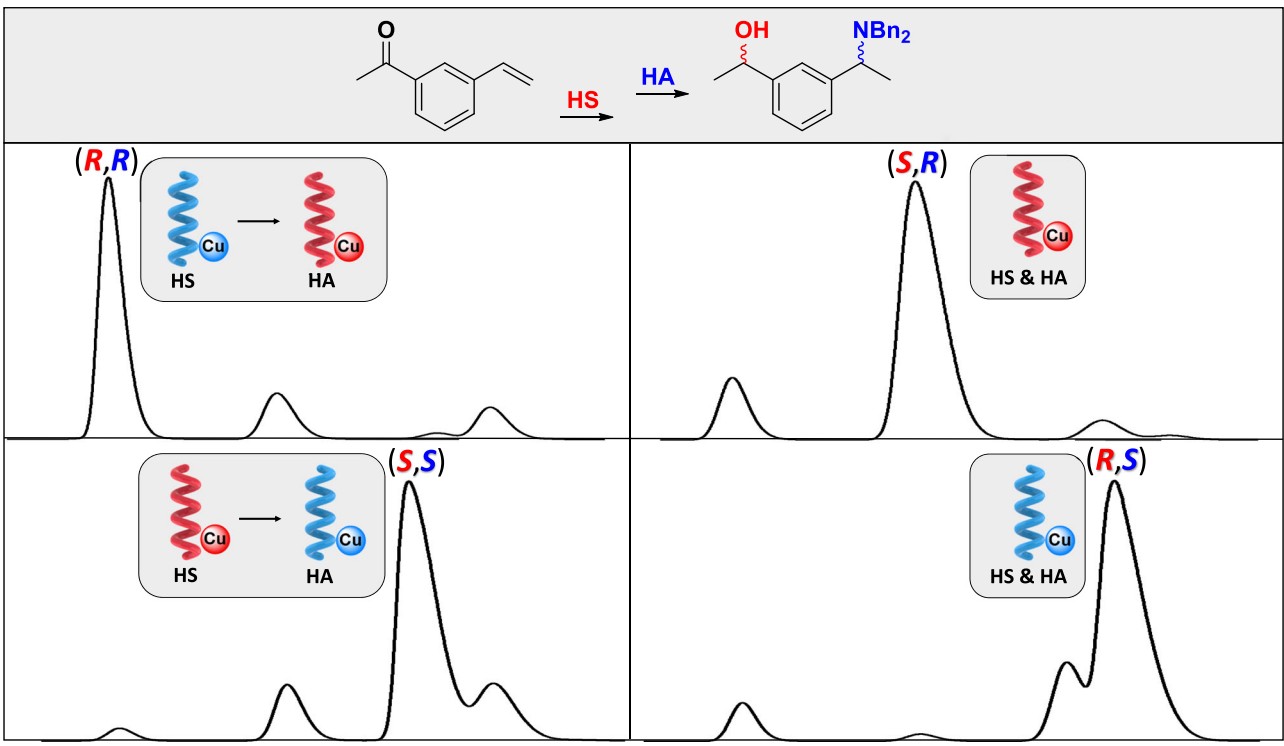

**Fig. 3 | Asymmetric stereodivergent catalysis with a stereochemically switchable BTA helical catalyst.** HPLC traces under the most selective conditions employing either the BTA helical catalyst with a single handedness for both hydrosilylation (HS) and hydroamination (HA) reactions (right, entries 1–2 of Table 1) or the BTA helical catalyst with opposite handedness for the HS and HA reactions (left, entries 3–4 of Table 2).

measurements (HRMS) were obtained on TQ R30-10 HRMS spectrometer by ESI+ ionization and are reported in m/z for the major signal. FT-IR analysis was performed on a Bruker Tensor 27 spectrometer in ATR (diamond probe).

## HPLC analyses
**APnol** stereoisomers have been isolated in enantiopure form by preparative HPLC (Supplementary Figs. 12–15, Supplementary Table 5). The following material has been used for HPLC analyses of catalytic experiments: Chiralpak IG-3 column (0.46 cm × 25 cm) bought from Chiral Technologies®, Waters 2487 Dual λ Absorbance Detector, with the following conditions: heptane/EtOH 98/2, 0.5 mL/min, 230 nm, 293 K. Retention times: between 19 and 26 min for **APnol**, between 33 and 37 min for **VPnol**. Ee, dr, ee1 and ee2 values are extracted from the HPLC traces as indicated in Supplementary Fig. 20. Error bars: when the main stereoisomer is (**R,R**)-**APnol** or (**S,R**)-**APnol**, the uncertainty is ±1% for the area of the main HPLC signal, ±1% for ee, ±0.2 for dr, ±1% for ee1 and ±1% for ee2 whilst when the main stereoisomer is (**S,S**)-**APnol** or (**R,S**)-**APnol**, the uncertainty is ±2% for the area of the main HPLC signal, ±1% for ee, ±0.4 for dr, ±3% for ee1, and ±1% for ee2 due to overlap of the signals. The uncertainty of ±2% has been estimated by deconvolution of the signals corresponding to the (S,S) and (R,S) stereoisomers of **APnol** in the case where these signals have significant different intensities. For the (R,R) and (S,R) stereoisomers, an error bar of ±1% is set relatively to the intrinsic uncertainty of the HPLC analysis. For the determination of the absolute and relative configurations of **APnol** (catalytic reactions performed with (S)- and (R)-DTBM-SEPHOS and MM/MD calculations): see Supplementary Figs. 16–19.

## CD analyses
Circular dichroism (CD) measurements were performed on a Jasco J-1500 spectrometer equipped with a Peltier thermostated cell holder and Xe laser. CD spectra were recorded at 313 K (except otherwise stated) with the following parameters: 50 nm.min⁻¹ sweep rate,

0.05 nm data pitch, 2.0 nm bandwidth, and between 400 and 275 nm. The solutions were placed into cylindrical spectrosil quartz cell of 0.05 mm pathlength (Starna® 31/Q/0.05). Toluene and cell contributions at the same temperature were subtracted from the obtained signals. CD analyses at different temperatures (Supplementary Fig. 6b) have been performed upon heating the solution containing the supramolecular tetrapolymer from 293 K to 313 K (heating rate: 2 K.min⁻¹).

## ITC analyses
ITC data (Supplementary Fig. 2) were recorded on a Microcal VP-ITC apparatus at the desired temperature, injecting a toluene solution containing **BTA P** (2.8 mM), **a-BTA** (2.8 mM), (**S**)-**BTA** (1.4 mM) corresponding to a total BTA concentration of 7.0 mM. into neat toluene. Injections of 5 μL over 10 seconds were performed every 480 seconds at a stirring rate of 300 rpm.

## MM/MD calculations
**APnol** stereoisomers were built and their conformations were generated using molecular mechanics (MM) and molecular dynamics (MD) methods implemented in the Materials Studio 6.0 modelling package [BIOVIA, Dassault Systèmes, Biovia Materials Studio, 6.0, San Diego: Dassault Systèmes, 2011]. The molecules were modelled without explicit solvent, using a distance-dependent dielectric constant. The atomic charges were assigned from the Polymer Consistent Force Field (PCFF)[75,76]. and a long-range interaction cutoff was set to 14 Å with a spline width of 1 Å. As a force field, Dreiding[77] was modified to properly reproduce van der Waals interactions of hydrogen atoms of atom type H_ (R0 _reduced from 3.195 Å to 2.83 Å)[78].

## Selected catalytic experiments
Note that the excess of DMMS (an eye-irritant chemical) is quenched with ammonium fluoride to avoid evaporation following the procedure described in reference[65].

**Control of the selectivity of the reaction by the catalyst.** catalytic experiments have been performed with a racemic mixture of (*S*)- and (*R*)-DTBM-SEPHOS or by a BTA helical catalyst incorporating an equimolar mixture of (*S*)-BTA and (*R*)-BTA, see Supplementary Figs. 10–11.

**Optimized conditions with helical BTA catalyst [concomitant process].** An oven-dried 50 mL Schlenk tube was loaded with **BTA P** (108.5 mg, 0.113 mmol, 10.0 mol%), Cu(OAc)$_2$ (10.3 mg, 0.056 mmol, 5.0 mol%), P(3,5-(CF$_3$)$_2$-C$_6$H$_3$)$_3$ (38 mg, 0.056 mmol, 5 mol%) and anhydrous THF (2.5 mL), and the mixture was stirred for 10 minutes. The solvent was then removed under vacuum and the tube was kept in vacuum (10$^{-3}$ mbar) for 1 hour. (*S*)-BTA (59.5 mg, 0.056 mmol, 5 mol%), **a-BTA** (123 mg, 0.113 mmol, 10.0 mol%) and anhydrous toluene (10 mL) were added to the tube and the mixture was briefly heated to reflux and stirred for 10 minutes at room temperature. **VPnone** (164.5 mg, 1.13 mmol, 100 mol%) and **Amine-tBu** (600 mg, 2.02 mmol, 180 mol%) were added to the tube. The Schlenk tube was sealed with a rubber septum, then evacuated and backfilled with N$_2$ three times. An atmosphere of nitrogen was maintained during the following addition. The reaction mixture was stirred and heated to 313 K, prior to the addition of DMMS (598 mg, 695 μL, 5.63 mmol, 500 mol%) via syringe. After overnight, the reaction mixture was cooled down to room temperature. A saturated solution of NH$_4$F (10 mL) in MeOH was added and the mixture was stirred until it became transparent. Then a saturated aqueous solution of Na$_2$CO$_3$ (10 mL) as well as EtOAc (5 mL) were added. The phases were separated and the aqueous layer was extracted with EtOAc (2×5 mL). The organic phases were collected and evaporated under vacuum. The crude product was then purified by flash column chromatography, eluting with petroleum ether/dichloromethane (gradient from 100/0 to 30/70), to yield **APnol** as a faint yellow oil (212 mg, 51% yield). The $^1$H NMR spectrum and the HPLC trace of **APnol** are shown in Supplementary Fig. 41 and Supplementary Fig. 29, respectively (also Fig. 3 for the HPLC trace). Catalysis with (*R*)-BTA instead of (*S*)-BTA was conducted following the same procedure, yielding **APnol** in 48% yield. The $^1$H NMR spectrum and the HPLC trace of **APnol** are shown in Supplementary Fig. 41 and Supplementary Fig. 30, respectively (also Fig. 3 for the HPLC trace).

**Optimized conditions with helical BTA catalyst [sequential process]**

An oven-dried reaction tube was loaded with **BTA P** (21.7 mg, 22.5 μmol, 10.0 mol%), Cu(OAc)$_2$ (2.1 mg, 11.3 μmol, 5.0 mol%), P(3,5-(CF$_3$)$_2$-C$_6$H$_3$)$_3$ (7.6 mg, 11.3 μmol, 5 mol%) and anhydrous THF (500 μL), was stirred for 10 minutes on a shaking machine. The solvent was then removed under vacuum and the tube was kept in vacuum (10$^{-3}$ mbar) for 1 hour. (*S*)-BTA (11.9 mg, 11.3 μmol, 5 mol%), **a-BTA** (12.3 mg, 11.25 μmol, 5.0 mol%) and anhydrous toluene (2 mL) were added to the tube and the mixture was briefly heated to reflux and stirred for 10 minutes at room temperature. **VPnone** (32.9 mg, 225 μmol, 100 mol%) was added to the tube. The reaction mixture was heated to 313 K and DMMS (140 μL, 1125 μmol, 500 mol%) was added. The reaction mixture was stirred for 15 minutes at 313 K. A solution of (*R*)-BTA (35.7 mg, 33.9 μmol, 15 mol%) in toluene (0.2 mL) was then added and the mixture was allowed to stir for additional 3 minutes at 313 K. Then **Amine-tBu** (120 mg, 405 μmol, 180 mol%) as well as DMMS (140 μL, 1125 μmol, 500 mol%) were added successively. After 2 hours, the reaction mixture was cooled down to room temperature. A saturated solution of NH$_4$F (2 mL) in MeOH was added to quench the mixture and the mixture was stirred until it became transparent. Then a saturated aqueous solution of Na$_2$CO$_3$ (2 mL) as well as EtOAc (1 mL) were added. The phases were separated and the aqueous layer was extracted with EtOAc (2×1 mL). 1,3,5-trimethoxybenzene (37.8 mg, 225 mol %) was added to the combined organic phases and the NMR yield was established after evaporation of the solvents. The mixture was then filtered over silica plug and eluted with dichloromethane prior to HPLC analysis. The $^1$H NMR spectrum and the HPLC trace of **APnol** are shown in Supplementary Fig. 8 and Supplementary Fig. 35 (also Fig. 2), respectively. Catalysis starting with (*R*)-BTA and with addition of (*S*)-BTA after the HS step (Table 2, entry 4) was conducted following the same procedure. The $^1$H NMR spectrum and the HPLC trace of **APnol** are shown in Supplementary Fig. 8 and Supplementary Fig. 36 (also Fig. 3), respectively.

For additional catalytic procedures, see the dedicated parts in the Supplementary Information.

### Notes quoted in the manuscript

Note 1: Throughout this paper, in the concomitant process, HS and HA reactions are initiated at the same time but the HA reaction is slower than the HS reaction (see Supplementary Fig. 3).

Note 2: The calculations are made by considering that conversion occurs only before or after the switch, i.e. with catalysts displaying opposite enantioselectivities. These values are actually lower limit values if one considers that conversion occurs also during the stereochemical switch, e.g. with a racemic catalyst.

Note 3: We attribute the discrepancy between the time of stereochemical switch probed by CD spectroscopy (Fig. 2) and that deduced from these catalytic experiments by a slower dynamicity of the supramolecular helices embedding the catalytic active species/resting states versus the pre-catalytic species. Whilst related supramolecular BTA terpolymers with copper acetate complexes anchored at their periphery have been found to be well-soluble single helices (see reference[56]), it can be surmised that the generation of hydride species, that tend to bridge several copper atoms, may generate aggregated supramolecular helices through copper crosslinks that are expected to be less soluble and less dynamic under our experimental conditions[59,64]. This might also explain why the rate of the stereochemical switch in the present case (on the timescale of minutes) is slower than in our previous study dealing with HS only and for which the switch occurred on the timescale of seconds[17].

### Data availability

The synthetic, analytical and modelling data generated in this study are all provided in the Supplementary Information. Raw data corresponding to CD spectra analyses of Fig. 2 have been deposited in the FigShare database (https://doi.org/10.6084/m9.figshare.25592565)[79].

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

## Acknowledgements

This work was supported by the China Scholarship Council (CSC-202106650009, PhD grant of R.C.), by the French Agence Nationale de la Recherche (project ANR–17-CE07-0002 AbsoluCat, PhD grant of A.H., PDRA grant of P.A.) and by the Consejo Nacional de Ciencia y Tecnologia (CONACYT, PDRA grant of M.A.M.-A.). Omar Khaled and Ludovic Dubreucq (IPCM, Sorbonne Université) are acknowledged for assistance with chiral HPLC experiments and preparation of starting materials, respectively. Gilles Clodic (MS3U, Sorbonne Université) is acknowledged for the ESI analysis of catalytic samples. The GDR 3712 Chirafun is acknowledged for allowing a collaborative network between the partners of this project.

## Author contributions

M.R. and L.B. designed the project. L.B. performed the ITC analyses. R.C. performed the optimization of the catalytic conditions for the cascade reaction (concomitant and sequential conditions), the ¹H NMR

monitoring of the catalytic reactions, CD analyses and the catalytic reactions on ca. 1 mmol scale. A.H. conducted catalytic experiments that led to the selection of the substrate, the preparation of **APnol** stereoisomers for isolation by preparative HPLC and initial conditions for the concomitant process. P.A. and M.A.M.-A. conducted preliminary catalytic and CD experiments, respectively, for the sequential process. N.V. performed the separation of the **APnol** stereoisomers by preparative HPLC and their chiroptical analyses. R.M. performed $^1$H NMR analyses of the catalytic mixtures. P.B. provided MM/MD calculations for the determination of the absolute and relative configurations of **APnol**. M.R. and R.C. prepared the overall manuscript, including the Figs. All authors contributed to the preparation of the manuscript by commenting and discussing the manuscript. The overall project was supervised by M.R.

## Competing interests

The authors declare no competing interests.
