## [Peer Review File · Nature Communications]

REVIEWER COMMENTS

Reviewer #1 (Remarks to the Author):

In this manuscript, four stereoisomers of an amino alcohol are accessed in one-pot, which avoids the purification process. The approach is made possible by the incorporation of an achiral benzene-1,3,5-tricarboxamide (BTA) phosphine ligand in a supramolecular polymer for which the optical purity and handedness are controlled by an enantiopure BTA monomer. The direction of the asymmetric reaction is directly related to the handedness of the helical polymer, which can be switched by selecting the main BTA enantiomer in the polymers at each step of the reaction, leading to predictable configuration of two stereogenic centers. The stereochemical switch occurs through the addition of an excess of the BTA enantiomer acting as a chiral chemical trigger that inverts the stereochemical preference of the copper catalyst. The achiral BTA additive present in the supramolecular terpolymer, as well as the secondary phosphine ligand, improve the selectivity and yield of the reaction, respectively. This work demonstrates the feasibility to select one major amongst four possible stereoisomers of an amino alcohol by applying the supramolecular helical catalyst in either concomitant or sequential, hydrosilylation and hydroamination reactions. This work seems interesting and can be acceptable for publication in Nature Communications. Some suggestions and concerns are listed as follows.

1. The stability of the supramolecular helical catalysts is suggested to describe in the manuscript. Is it tolerant to temperature and solvents? The CD and UV-vis spectra of the supramolecular polymers are suggested to measure in different solvents at different temperatures and at different concentrations.
2. For its chiral catalysis, the loading of BTA catalyst is large, and whether this kind of catalyst is easy to recover or can be recycled experiments?
3. For the helix inversion by adding the enantiomer inducer, it is either continued growth on terminal of the original supramolecular assemblies or disassembly followed by reorganization, or self-assembly of the newly added monomers alone. The possible mechanism of the helix interconversion is encouraged to provide.
4. The phosphine-coordinated copper is located at the peripheral of the helix, so how the helix affects the selectivities of the reactions?
5. Some papers related to this work closely are encouraged to cite, such as Nature Communications 2023, 14, 7287. Angew. Chem. Int. Ed. 2023, 62, e202310105. Nature Communications 2023, 14, 566.

Reviewer #2 (Remarks to the Author):

The manuscript "Switchable supramolecular helices for asymmetric stereodivergent catalysis" submitted by Raynal and coworkers shows an elegant method to predominantly obtain 1 out of 4 possible stereoisomers by using a switchable supramolecular catalyst that 1) can catalyze 2 different reactions, a

hydrosilylation and a hydroamination, and 2) can switch helical sense by addition of the opposite enantiomer that acts as a sergeant hereby switching the enantioselectivity of the reaction. Previous work by this group has shown that the chiral BTA-based helices with metals embedded can perform a variety of catalytic reactions well and with reasonable (albeit never excellent) enantioselectivities. The authors now redesign the system to perform 2 catalytic reactions either at the same time or sequentially, and control the helical sense of the carrier for the Cu catalyst such that the outcome of the reaction is also biased. Although switchable asymmetric catalysts attracted a lot of attention in recent years, as far as I know, there is nothing similar as the authors now work out in this manuscript. Although the selectivities are not really high for the asymmetric catalysis field (which likely is a limitation of the scaffold they choose to work with), the results are very convincing and the approach is highly original.

The noteworthy result of this work is the fact that 1 catalyst can catalyse 2 different reactions in a stereoselective manner and by switching the helical sense of the scaffold supramolecular polymer, easily done by adding opposite enantiomer, one can select which stereoisomer can be biased in the second step. All this is done in a one pot procedure. While the selectivities of the products are too low to be of immediate practical relevance, this work is important. It shows that switching helical sense of a supramolecular scaffold can be applied in a broader context than for only 1 specific reaction, and that hereby more stereocentres can be controlled on one substrate. This makes the work very original. The addition of the HPLC traces is also very convincing to see what happens to the stereochemistry of the product by selecting reaction conditions. All in all, the results are well presented and well discussed making the manuscript easy to read as well as very insightful. The only weak part of the manuscript is the fact that no pure stereoisomers can ever be obtained, but this is a limitation of the selected scaffold system. On the other hand, this also provides opportunities to develop other and better scaffolds that are more or less dynamic and hereby provide higher stereoselectivities. The authors provide a detailed ESI and detailed descriptions of how the reactions have been performed, and provide crude data to support their claims (LC traces, NMR spectra). To my opinion all conclusions are well supported by the experiments, also because the authors have quite some experience and detailed understanding for similar systems.

I do have a few comments and questions the authors may want to address

1. The HPLC traces in the Table are really nice and insightful, but it would be really great if the different configurations / peaks could somehow be colour coded so it is immediately evident which peak belongs to which configuration
2. SS and RS peaks are not baseline separated in the LC traces, and it seems that the integration was done (at least from the ESI) by simple marking of the middle boundary. Or were the peaks deconvoluted to assess more reliably the peak areas?
3. In the conclusions the authors state that “xxx avoids the purification process required xxx”. However, in their case, no pure stereoisomers can be obtained, so I would suggest to make the conclusion a bit more careful: in the ideal case, if ever enzyme-like enantioselectivities could be reached, no purification will be needed but with this particular system, this is not the case, and if a single diastereomer is required purification needs to be done.

Reviewer #3 (Remarks to the Author):

The manuscript submitted by M. Raynal et al. describes a consequent extension of previously published work on the application of copper complexed by helically chiral supramolecular BTA stacks for asymmetric hydroaminations (HA) and hydrosilylations (HS). It is a really impressive example for the exploitation of a configurationally dynamic macromolecular system (if one looks at it as if it were a polymer) to control the absolute and relative configuration of a product displaying two stereogenic centers (two pairs of diastereomeric enantiomers). The rationale behind is based on the "sergeant and soldier" as well as the "majority rules (MR)" principle which allows for helicity control by only minor amounts of a chiral BTA which even has not to be enantiomerically pure (diluted MR). The enantiomers of diastereomer 1 can be accessed by a one-pot concomitant reaction with either the M- or P-helical Cu-complexed BTA assembly and the enantiomers of diastereomer 2 by a reaction sequence starting with the HS followed by the HA-reaction after a helical inversion of the BTA stack. This helical inversion was effected by simply adding "the other" chiral BTA comonomer (diluted MR).

Although the diastereoselectivities achieved are not that spectacular (to say the least), the enantioselectivities are. Given the fact that this way to control the outcome of a stereoselective catalysis is really rare (I'm not aware of another one) these achievements are definitely more than just a beginning. I'm convinced that this kind of dynamic control of stereoselective reactions is a milestone on the way to "next generation of asymmetric catalysis".

Nonetheless I like to make some comments:

1) From the stereochemical outcome of the reactions it is obvious that the helicity of the stacks dominates the selectivity, not so the chiral sidechains (as can be seen from missing "matched" and "mismatched" combinations between helical- and central chirality. The central role of the helical chirality in an asymmetric catalysis has been shown in its "purest" (helicity only - no chirality element in the repeating units) form by the group of Reggelin in 2002 (*Angew. Chem., Int. Ed.* 2002, 41, 1614-1617). Furthermore, the same group in 2004 showed a successful application of the "sergeant and soldier" principle in asymmetric hydrogenations using helically chiral, dynamic polyisocyanates (*Proc. Natl. Acad. Sci. U.S.A.* 2004, 101, 5461-5466). Both pioneering papers should be cited.

2) A disadvantage of the current "chirality-switching procedure" by the addition of "the other" chiral BTA is obviously that this cannot be done more than one or maybe two times (the ee of the chiral BTA will approach 0). It would be better to reversibly switch helicity by irradiation of light. In the case of CPL it may even be possible to omit chiral BTAs at all - IF (a big IF!) the induced helicity in the BTA stack is configurationally stable at the time scale of the subsequent reactions.

Fazit:

An impressive piece of work that should be published as is (after addition of the two references, mentioned above).

Reviewer 1: This work demonstrates the feasibility to select one major amongst four possible stereoisomers of an amino alcohol by applying the supramolecular helical catalyst in either concomitant or sequential, hydrosilylation and hydroamination reactions. This work seems interesting and can be acceptable for publication in Nature Communications. Some suggestions and concerns are listed as follows.

Reply: Thank you very much for your strong support and comments.

1. The stability of the supramolecular helical catalysts is suggest to describe in the manuscript. Is it tolerate to temperature and solvents? The CD and UV-vis spectra of the supramolecular polymers are suggest to measure in different solvents at different temperatures and at different concentration.

Reply: As the supramolecular helical catalysts are built on hydrogen bonding interactions, they are indeed quite sensitive to the operatory conditions. Isothermal Titration Calorimetry (ITC) is a particularly suitable technique to probe the stability of these dynamic polymers (Langmuir, 2018, 14176). We have now added ITC analyses of the initial catalytic system performed at 293 K and 313 K (Figure S1). The following sentence has been added in the main text: "Isothermal Titration Calorimetry (ITC) experiments reveal that the supramolecular helical catalyst is stable above 0.4 mM at 293 K and 1.3 mM at 313 K in toluene: the concentration in BTA P has been set to 11 mM to ensure that helices are maintained during the full catalytic process conducted at 313 K (Figure S1)." Our previous studies (refs [17] and [56]) indicated that toluene is a suitable solvent to maintain a good balance between solubility, stability and dynamicity. Obviously, our catalytic system is less amenable to optimization by solvent screening than conventional or helical covalent catalysts. Anyway, we have now added the catalytic results obtained with a small set of aromatic solvents (Table S2). We observe a slight

improvement in the ee of the hydroamination step when the reaction is performed in trifluorotoluene, but the overall yield and selectivity are barely affected by the nature of the solvent. We have added the following sentence in the main text: "Other aromatic solvents were also probed but they showed no obvious advantage relatively to toluene (Table S2)."

2. For its chiral catalysis, the loading of BTA catalyst is large, and whether this kind of catalyst is easy to recover or can be recycled experiments?

Reply: Copper hydride catalysis requires quenching (NH_4F in MeOH) at the end of the experiment to cleave the OSi bond and neutralize the excess of silane reagent. It is probable that the phosphine copper organometallic complex is oxidized under these conditions and thus not suitable anymore to catalysis. BTA monomers are easily separated from the catalytic products by either precipitation in MeCN or purification by flash chromatography as can be seen by the fact that **APnol** is isolated pure (pages S40-S41 and S51). Organocatalytic systems would be more prone to recyclability, we will test this possibility in the future.

3. For the helix inversion by adding the enantiomer inducer, it is either continued growth on terminal of the original supramolecular assemblies or disassembly followed by reorganization, or self-assembly of the newly added monomers alone. The possible mechanism of the helix interconversion is encouraged to provide.

Reply: BTA assemblies are known to be highly dynamic in solution with reorganization of the monomers occurring rapidly upon many possible mechanisms, one possibility being that monomers exchange homogeneously along the polymer backbone (Science, 2014, 491). We have now added CD analyses of the catalytic system after stereochemical switch; either upon addition of the chiral BTA or upon cooling from the monomeric state. The CD spectra are virtually identical (Figure S5). We have thus added the following sentence in the main text: "CD spectra of the supramolecular tetrapolymer upon formation from preexisting terpolymer or after cooling from the monomeric state are virtually identical thus indicating that the thermodynamic state is reached after switching of the helix handedness (Figure S5)." In other words, the system is sufficiently dynamic at 313 K to integrate the added chiral BTA monomers into the helices and form the most stable helical coassemblies. The distribution of the monomers is expected to be random given their similar structure but the exact distribution of the monomers in the helical coassemblies cannot be fully ascertained at this point.

Please also note that we have now provided better-resolved CD spectra in Figure 1 since we noticed that the absorbance was too high on the previously-reported CD spectra. The difference in intensity comes to the fact that now the analyses have been performed in a 0.05 mm cell (instead of 0.1 mm cell for the previous data). The CD spectra before and after stereochemical switch are now perfectly mirror images from each other.

4. The phosphine-coordinated copper is located at the peripheral of the helix, so how the helix affect the selectivities of the reactions?

Reply: The phosphine-coordinated copper complex, even though being intrinsically achiral, is located in the chiral environment provided by the helical coassemblies as demonstrated by the induced CD band detected in the CD analyses. Our previous report on a related catalytic system demonstrates that the role of the chiral monomer is mainly to bias the helix handedness (reference [57]). It is thus assumed that the phosphine copper complex may adopt a chiral conformation in the helical coassemblies as postulated for a related rhodium complex in our previous study (reference [53]).

5. Some papers related to this work closely are encouraged to cite, such as Nature Communications 2023, 14, 7287. Angew. Chem. Int. Ed. 2023, 62, e202310105. Nature Communications 2023, 14, 566.

Reply: References dealing with covalent helical catalysts bearing intrinsically achiral catalytic centres (references [38-50]), synergistic catalysis with a helical polymer bearing chiral catalytic sites ([51]), and chiral switchable polymerization catalysts (reference [14]) have been added to the manuscript and this includes the references suggested by the referee.

Reviewer 2: To my opinion all conclusions are well supported by the experiments, also because the authors have quite some experience and detailed understanding for similar systems.

Reply: Thank you very much for your strong support and comments.

1. The HPLC traces in the Table are really nice and insightful, but it would be really great if the different configurations / peaks could somehow be colour coded so it is immediately evident which peak belongs to which configuration.

Reply: The configuration of each stereoisomer in their order of elution has been provided in the header line of the two catalytic tables present in the main text.

2. SS and RS peaks are not baseline separated in the LC traces, and it seems that the integration was done (at least from the ESI) by simple marking of the middle boundary. Or were the peaks deconvoluted to assess more reliably the peak areas?

Reply: The (S,S) and (R,S) stereoisomers of **APnol** are indeed not perfectly separated. We have compared HPLC areas of deconvoluted and non-deconvoluted peaks in the worst case for which signals have very different intensities.

Without deconvolution:

With deconvolution:

As you can see, the difference in %area is below 2%. Accordingly, HPLC traces have not been deconvoluted but we have indicated slighter higher uncertainties for these two stereoisomers as indicated in the captions of Tables 1 and 2 and in the method section: "(...) when the main stereoisomer is (S,S)-**APnol** or (R,S)-**APnol**, the uncertainty is $\pm 2\%$ for the area of the main HPLC signal, $\pm 1\%$ for ee, ± 0.4 for dr, $\pm 3\%$ for ee1, and $\pm 1\%$ for ee2 due to overlap of the signals. The uncertainty of $\pm 2\%$ has been estimated by deconvolution of the signals corresponding to the (S,S) and (R,S) stereoisomers of **APnol** in the case where these signals have significant different intensities."

3. In the conclusions the authors state that "xxxx avoids the purification process required xxxx". However, in their case, no pure stereoisomers can be obtained, so I would suggest to make the conclusion a bit more carefull: in the ideal case, if ever enzyme-like enantioselectivities could be reached, no purification will be needed but with this particular system, this is not the case, and if a single diastereomer is required purification needs to be done.

Reply: We fully agree that the presented concept allows to avoid the purification of the reaction intermediate, not the final purification step needed to separate the amino alcohol(s) from the catalyst and reactants. The main text has been modified as follows: "The four stereoisomers of an amino alcohol are accessible one-pot, representing a totally new approach which notably avoids isolation of the product intermediate from the initial catalytic system that is required when two enantiomers of a non-switchable catalyst are needed for each step of a sequential catalytic process."

Reviewer 3: Given the fact that this way to control the outcome of a stereoselective catalysis is really rare (I'm not aware of another one) these achievements are definitely more than just a beginning. I'm convinced that this kind of dynamic control of stereoselective reactions is a milestone on the way to "next generation of asymmetric catalysis".

Reply: Thank you very much for your strong support and comments.

1) From the stereochemical outcome of the reactions it is obvious that the helicity of the stacks dominates the selectivity, not so the chiral sidechains (as can be seen from missing "matched" and "mismatched" combinations between helical- and central chirality. The central role of the helical chirality in an asymmetric catalysis has been shown in its "purest" (helicity only - no chirality element in the repeating units) form by the group of Reggelin in 2002 (Angew. Chem., Int. Ed. 2002, 41, 1614-1617. Furthermore, the same group in 2004 showed a successful application of the "sergeant and soldier" principle in asymmetric hydrogenations using helically chiral, dynamic polyisocyanates (Proc. Natl. Acad. Sci. U.S.A. 2004, 101, 5461-5466. Both pioneering papers should be cited.

Reply: References dealing with covalent helical catalysts bearing intrinsically achiral catalytic centres have been added (references [38-50]) and those dealing with helical catalysts devoid of stereogenic centers (including the two references suggested by the referee) have been particularly stressed as references [38-40].

2) A disadvantage of the current "chirality-switching procedure" by the addition of "the other" chiral BTA is obviously that this cannot be done more than one or maybe two times (the ee of the chiral BTA will approach 0). It would be better to reversibly switch helicity by irradiation of light. In the case of CPL it may even be possible to omit chiral BTAs at all - IF (a big IF!) the induced helicity in the BTA stack is configurationally stable at the time scale of the subsequent reactions.

Reply: In overall, we agree with the referee that a limitation of the current approach is the necessity to add in situ an excess of a chiral monomer. However, this can be made in principle several times (e.g. 0% ee → -20% ee → +33% ee → -50% ee), the only limitation is that all the monomers must be integrated into the same supramolecular helices. We will test in the future the possibility to control the configuration of stereogenic centers >2. We fully agree that CPL would be the ideal candidate for that purpose!

Minor typos have also been considered upon revision.

REVIEWERS' COMMENTS

Reviewer #1 (Remarks to the Author):

The authors have revised the manuscript, and the concerns were properly addressed. Thus, I recommend its publication in Nature Communications.

Reviewer #2 (Remarks to the Author):

The revised manuscript "Switchable supramolecular helices for asymmetric stereodivergent catalysis" by Raynal and coworkers has addressed all my concerns as well as those of the other reviewers. As stated before, this is really nice and fundamental work, and I look forward to seeing the manuscript in print